# Optimal Sensor and Relay Nodes Power Scheduling for Remote State Estimation with Energy Constraint

**DOI:** 10.3390/s20041073

**Published:** 2020-02-16

**Authors:** Yufei Han, Mengqi Cui, Shaojun Liu

**Affiliations:** 1Department of Automation, University of Science and Technology of China, Auhui 230027, Hefei, China; hanyufei@mail.ustc.edu.cn; 2School of Energy and Power, Jiangsu University of Science and Technology, Zhenjiang 212003, Jiangsu, China; cuimengqijust@163.com

**Keywords:** state estimation, relay nodes, wireless sensor networks

## Abstract

We study the sensor and relay nodes’ power scheduling problem for the remote state estimation in a Wireless Sensor Network (WSN) with relay nodes over a finite period of time given limited communication energy. We also explain why the optimal infinite time and energy case does not exist. Previous work applied a predefined threshold for the error covariance gap of two contiguous nodes in the WSN to adjust the trade-off between energy consumption and estimation accuracy. However, instead of adjusting the trade-off, we employ an algorithm to find the optimal sensor and relay nodes’ scheduling strategy that achieves the smallest estimation error within the given energy limit under our model assumptions. Our core idea is to unify the sensor-to-relay-node way of error covariance update with the relay-node-to-relay-node way by converting the former way of the update into the latter, which enables us to compare the average error covariances of different scheduling sequences with analytical methods and thus finding the strategy with the minimal estimation error. Examples are utilized to demonstrate the feasibility of converting. Meanwhile, we prove the optimality of our scheduling algorithm. Finally, we use MATLAB to run our algorithm and compute the average estimation error covariance of the optimal strategy. By comparing the average error covariance of our strategy with other strategies, we find that the performance of our strategy is better than the others in the simulation.

## 1. Introduction

Recent years have witnessed a boom of WSNs [1]. In the application of WSNs, sensors’ remote state estimation plays a key role.To process the estimation, the sensors in WSNs collect the information of physical phenomena such as temperature and humidity, record them as the “state”, run a preprocess for the state, and then send the state information as data packets to a remote estimator via a wireless network. Each procedure of this process has received extensive research from different perspectives  [1,2,3,4,5,6,7,8,9,10,11,12,13,14].

In this work, we study the optimal sending strategy for the sensor and the relay nodes in a WSN. Different from most previous works that designed sending strategies for the WSN with only a single sensor [2,3,4,5,6,7,8], in this work, the strategy is designed for a WSN with one sensor, one remote estimator, and extra relay nodes in between. The signal sent from the sensor no longer reaches the estimator directly, but is passed on by each relay node before finally reaching the estimator. Therefore, we assume that the sensor and the relay nodes all had the same energy storage for sending signals. The wireless channels links the sensor, the relay nodes, and the estimator. While being passed on, the signals sent by the high power are passed on by the relay nodes successfully, whereas the signals sent by the low power suffer packet drops. Then, the optimal strategy minimizes the error covariance of the remote estimator given the fixed amount of energy. Before demonstrating the strategy, we introduce the most recent works that are related to similar topics and explain the reasons we study this model.

### 1.1. Motivation and Background

To stop a high packet drop rate from deteriorating the performance of the remote estimator, relay nodes are added between the sensor and the remote estimator [11,12,13,14,15,16,17,18,19,20,21,22]. Relay nodes help pass on signals from the sensor to the final estimator and thus break the long wireless communication path into shorter sections that have a smaller packet drop rate. This assertion was motivated by the indication of M.Holland et al. [23] that the larger the distance between the sensor and the estimator, the larger the packet drop rate would be. A recent study regarding state estimation with relay nodes [24] emphasized the stability of the system, whereas we focused on finding the optimal strategy.

Our model extended Shi’s model [2]. Shi et al. gave the optimal sending strategy for the WSN that transmitted signals from the sensor to the estimator directly. In [2], the authors revealed that for a finite-time and finite-energy sensor data scheduling problem, sending every signal with a high power that ensures a successful transmission to the remote estimator does not make the best use of the energy. Under the assumption that sending low power signals will bring packet drops, they deduced that the optimal choice was to send high power signals mixed with low power signals and make them distribute as uniformly as possible. In this way, the sending strategy minimizes the average estimation error covariance of the remote estimator.

As was proposed in Asshad’s survey [22], in the WSNs with relay nodes, nodes with limited energy need to have a power optimization mechanism. To find the mechanism, in recent years, many papers considered methods concerning state estimation and sensor scheduling. Y. Yao et al. [25] studied the method for sensors’ position estimation in an ad hoc WSN with large numbers of sensor nodes and addressed an energy-efficient method. The authors made full use of distance estimation. PM. Daflapurkar et al. [26] came up with an energy-saving algorithm for cluster routing in the WSN. It focused on designing routing paths between nodes rather than node-to-node sensor scheduling strategies. P. Cheng et al. [24] built three node-to-node data forwarding strategies for state estimation in the WSN with relay nodes. To adjust the trade-off between the energy consumption and the state estimation accuracy, the authors designed an online strategy that compared the error covariance gap between two adjacent nodes to a predefined threshold. R. Zhu et al. [27] used a Markov chain to characterize the delay of the relay nodes of a multi-hop WSN and obtained the necessary and sufficient condition under which the estimation error covariance was stable. L. Yao et al. [28] investigated optimal scheduling for transmission in the WSN with one relay node that was placed in the feedback loop. Apart from the theoretical analysis, F.A. Aderohunmu et al. [14], M. Maggiorotti et al. [13], and M.Rossi et al. [12] also developed the real case applications for the power scheduling problem.

Compared to previous works, our model adds relay nodes in the model of [2], and the way the relay nodes update their state is the same as the local processing and forwarding strategy in [24]. However, different from the work of P. Cheng et al.in [24], we design an offline strategy that controls whether to send a signal or not for each relay node at each time step. Hence instead of a predefined threshold that adjusts the trade-off of energy and state estimation performance, we find an optimal offline sensor scheduling algorithm that achieves the smallest average estimation error covariance with limited energy. The detailed introduction of the problem we solve and the assumptions we adopt for our model are demonstrated as follows.

We considered the sensor power scheduling problem based on the relay system shown in Figure 1. The state information was preprocessed in the dashed line box, and the preprocessed data was sent to Relay Node 1 via a wireless channel that suffered packet drops. Then, the data packets were passed on from one relay node to the next through the wireless channel until they finally reached the remote estimator. For simplicity, we first analyzed the case when there were only two relay nodes in the system. Each relay node’s behavior was a combination of the sensor and the remote estimator. It received data packets as the estimator did, and sent the packets to the next node as the sensor does. As assumed previously, we only considered the energy distributed to send data and did not address the energy requirements for other functions such as detecting signals in the air, receiving the signals, and preprocessing them. Furthermore, the total amount of energy every relay node used for sending signals was equivalent to that of the sensor. Our goal was to find the optimal sending strategy to minimize the accumulation of the average estimation error of the remote estimator. To achieve this goal with the model and the assumptions we made above, we developed a scheduling strategy called the “converted table” method. The challenge was that the error covariance of the remote estimator was the result of two kinds of updates: the sensor-to-relay-node kind and the relay-node-to-relay-node kind. We were able to find the optimal scheduling methods for both kinds of updates, respectively, but when they were combined to produce one error covariance for the estimator, it would be hard to compare two scheduling strategies by subtracting one’s covariance formula representation from another as L. Shi et al. [2] did. Therefore, we used the “converted table” to change the sensor-to-relay-node update to the relay-node-to-relay-node update. Then, with the majorization theory [29], we were able to prove that the overall optimal scheduling strategy for the sensor and relay nodes was the combination of the optimal sensor-to-relay-node scheduling with the optimal relay-node-to-relay-node scheduling. We also discussed the case when the total amount of energy and the time horizon were not infinite and explained that there would be no way for the states or the error estimations of the nodes to obtain an optimal strategy. Therefore, we only focused on the finite-time and finite-energy case and developed algorithms within the scope of our assumptions.

### 1.2. Main Contributions

We found an optimal scheduling algorithm for our model. We proposed an original method called the “converted table” method to obtain the optimal sending sequences for the sensor and the relay nodes. This method not only reduced the estimator’s average error covariance effectively, but also spared us from the cumbersome exhaustive computation for searching for the optimal strategy.We analyzed the reason why the infinite-time case did not exist in our model.We ran simulations for our algorithm and compared its sending sequences’ average error covariance with error generated by other sending strategies. The result of the comparison was empirical proof of the optimality of our algorithm.

### 1.3. Organization

The remainder of the paper is organized as follows. First, Section 2 introduces the optimal sensor and relay nodes’ data scheduling scheme for the finite-time and limited energy case. Furthermore, it gives the proof for the scheme’s optimality. Then, Section 3 explains the non-existence of the scheduling for the infinite-time and energy case. Next, Section 4 shows the simulation result, which compares our optimal strategy with other strategies. Conclusions and consideration for future work are added in Section 5.

### 1.4. Notations

R is the set of real numbers. $R^{n}$ is the set of *n*-dimensional vectors. $Z_{+}$ is the set of non-negative integers. $k\in Z_{+}$ is the time index. $S^{n}$ is the set of *n* by *n* matrices. For $X\in S^{n}$, X>0 (and X≥0) implies that *X* is positive definite (and positive semi-definite), and it is represented as $X\in S_{++}^{n}$ (and $X\in S_{+}^{n}$). $X^{\prime }$ indicates the transposition of the matrix *X*. $I=X^{0}$ implies the identity matrix. E[X] is the expectation of a random variable *X*, and E[X|Y] denotes the conditional expectation of *X* given the event Y. Pr[Y] indicates the probability of the event Y. Tr(X) indicates the trace of the matrix *X*. For functions $h:S_{+}^{n}\to S_{+}^{n}$, we let $h^{k}\left(X\right)=h\left(h^{k-1}\left(X\right)\right)$ with $h^{0}\left(X\right)=X$.

## 2. Optimal Sensor and Relay Nodes’ Power Scheduling

Our goal was to find the optimal sensor and relay nodes’ sending strategy for the system shown in Figure 1. To follow the precedent, we set the state information as $x_{k}$ and the preprocessed data ${\hat{x}}_{k}^{s}$. The dashed arrows imply the wireless communication paths the signals take. We will first demonstrate the sending strategy that optimizes the scheduling for the model without the relay nodes and then generalize this strategy to find the optimal scheduling for the model with the relay nodes.

### 2.1. The No-Relay-Node Case

This case is equivalent to the case that analyzes the transmission between the sensor and Relay Node 1. Assume that this model is built as a dynamic discrete linear time-invariant system shown as follows:(1)$$\begin{array}{c} \left\{\begin{array}{cc} x_{k+1} & =Ax_{k}+w_{k}\\ y_{k} & =Cx_{k}+v_{k} \end{array}\right. \end{array}$$
where $k\in Z_{+}$. $x_{k}\in R^{n}$ is the system state at time *k* and $y_{k}\in R^{m}$ is the sensor’s measurement of the system state. $w_{k}\in R^{n}$ and $v_{k}\in R^{m}$ are zero-mean i.i.d. Gaussian noises with covariances Q≥0 and R>0, respectively. Here, we assume that the pair (A,C) is observable and $\left(A,\sqrt{Q}\right)$ is controllable.

At time *k*, the sensor’s measurements of the system states are represented as $Y_{k}=\left\{y_{1},\cdots ,y_{k}\right\}$. Then, in the preprocessor shown in Figure 1, the sensor with sufficient computation ability uses $Y_{k}$ to generate a minimum mean squared error estimate (MMSE) ${\hat{x}}_{k}^{s}=E\left[x_{k}|Y_{k}\right]$ with the corresponding error covariance $P_{k}^{s}=E\left[\left(x_{k}-{\hat{x}}_{k}^{s}\right){\left(x_{k}-{\hat{x}}_{k}^{s}\right)}^{\prime }|Y_{k}\right]$. By the standard Kalman filtering [17], if we define the function *h*: $S_{+}^{n}\to S_{+}^{n}$ as $h\left(X\right)=AXA^{\prime }+Q$ and $\tilde{g}$: $S_{+}^{n}\to S_{+}^{n}$ as $\tilde{g}\left(X\right)=X-XC^{\prime }{\left[CXC^{\prime }+R\right]}^{-1}CX$ just like the definition shown in [3], then the estimation error covariance $P_{k}^{s}$ converges to a constant value $\bar{P}$ exponentially fast, and we denote it as $lim_{k\to \infty }P_{k}^{s}=\bar{P}$, where $\bar{P}$ is the unique positive semi-definite solution of $\tilde{g}\circ h\left(\bar{X}\right)=X.$ Therefore, for simplicity, we assume that for $\forall k\in Z_{+}$, $P_{k}^{s}=\bar{P}$. To generalize the model without relay nodes into the model with relay nodes in Figure 1, in the following, we mark the sensor as the zeroth node and Relay Node 1 the first node; thus, we use $P_{k,i}$ to represent the error covariance generated at the $i^{\mathrm{th}}$ node at time *k*, and ${\hat{x}}_{k}^{s}$ is also marked as ${\hat{x}}_{k,0}^{s}$. In the precedents where there are no relay nodes, the initial error covariance of the remote estimator is the same as the sensor’s. Without this initial error covariance, the remote estimator cannot update its estimation of the sensor locally according to its a priori knowledge of the sensor under a packet drop. Therefore, when there are relay nodes, we assume the initial error covariance of the relay nodes is the same as the estimator, which enables local updates, i.e., $P_{0,i}=P_{0,0}=\bar{P}$.

**Lemma** **1** **([8]).**
*The function $h(\bar{P})$ defined above has the following property:*
$$\bar{P}\leq h\left(\bar{P}\right)\leq h^{2}\left(\bar{P}\right)\leq \cdots \leq h^{k}\left(\bar{P}\right)\leq \cdots ,\forall k\in Z_{+}.$$


Note that for $\forall k\in Z_{+}$, we have $h^{k}\left(\bar{P}\right)\geq 0$, so $Tr[h^{k}\left(\bar{P}\right)]\geq 0.$

To calculate the average error covariance, first we define the scheduling variable of the $i^{\mathrm{th}}$ node $\lambda_{k,i}$ as follows:$$\lambda_{k,i}=\left\{\begin{array}{cc} 1, & \mathrm{if}\;a\;\mathrm{high}\;\mathrm{power}\;\mathrm{is}\;\mathrm{sent}\;\mathrm{at}\;\mathrm{time}\;k\\ 0, & \mathrm{if}\;a\;\mathrm{low}\;\mathrm{power}\;\mathrm{is}\;\mathrm{sent}\;\mathrm{at}\;\mathrm{time}\;k. \end{array}\right.$$

Let $T\in Z_{+}$ be the time horizon, then the $i^{\mathrm{th}}$ node’s scheduling sequence is $\theta_{i}=\left(\lambda_{1,i}\cdots ,\lambda_{T,i}\right)$. Besides, as was assumed in [2], we also assume that at time *k*,
$$\mathrm{Packet}\;\mathrm{drop}\;\mathrm{rate}\;\alpha_{k}=\left\{\begin{array}{c} 0,\;\mathrm{if}\;\lambda_{k,i}=1\\ \alpha ,\;\mathrm{if}\;\lambda_{k,i}=0. \end{array}\right.$$

Therefore at the Relay Node 1, that is the 1^th^ node’s state estimate ${\hat{x}}_{k,1}$ and the error covariance $P_{k,1}$ are updated as follows:(2)$$\begin{array}{c} \left({\hat{x}}_{k,1},\;P_{k,1}\right)=\left\{\begin{array}{cc} (A{\hat{x}}_{k-1,1},\;h\left(P_{k-1,1}\right)), & \mathrm{if}\;\mathrm{the}\;\mathrm{packet}\;\mathrm{drops}\\ ({\hat{x}}_{k}^{s},\;P_{k,1}), & \mathrm{if}\;\mathrm{the}\;\mathrm{packet}\;\mathrm{arrives}. \end{array}\right. \end{array}$$
where $P_{0,1}=P_{0,0}=\bar{P}$.

Thus, the average error covariance of the state updates in the following way [16]:(3)$$\begin{array}{cc} E[P_{k+1,0}] & =\left(1-\alpha_{k}\right)P_{0,0}+\alpha_{k}h\left(E\left[P_{k,0}\right]\right). \end{array}$$

Then, we use the trace of the average expected estimation error covariance to evaluate the system performance, just as the authors in [2,3,17,18] did, i.e.,
(4)$$\begin{array}{c} J_{\left[i\right]}\left(\theta \right)=1/T\sum_{k=1}^{T}E\left[P_{k,i}\left(\theta \right)\right]. \end{array}$$

The goal of the optimal sensor scheduling problem is to optimize the following problem:

**Problem** **1.**
$$\min_{\theta }Tr\left(J_{\left[1\right]}\left(\theta \right)\right)$$
$$s.t.\sum_{k=1}^{T}\lambda_{k,0}=m$$
*where m indicates the times that the high power is sent. Though our final goal was different, it was still necessary to work out this problem first. Problem 1 can be solved as a dual problem of Problem 3.1 from [18]. In that paper, the problem was proposed from the perspective of an attacker. However, it can be modified into the optimal sensor power scheduling. For clarity, we transformed the format of the solution of Problem 3.1 in [18] into the solution of our power scheduling problem shown as follows.*


**Theorem** **1.**
*(1) If m<(1/2)T, the optimal solution for Problem 1 is the following sequence of θ.*
$$\underset{l}{\underset{︸}{\left(0\cdots 0\right)}}\;\underset{mp+m+l-T}{\underset{︸}{\underset{p}{\underset{︸}{\left(1\;0\cdots 0\right)}}\;......\underset{p}{\underset{︸}{\left(1\;0\cdots 0\right)}}}}\;\underset{T-l-mp}{\underset{︸}{\underset{p+1}{\underset{︸}{\left(1\;0\cdots 0\right)}}\;......\;\underset{p+1}{\underset{︸}{\left(1\;0\cdots 0\right)}}}}$$
*where $p=\lceil \frac{T-l-m}{m}\rceil$ and l=p−1 or p (⌈x⌉ is defined as the smallest integer that is larger than x).*

*(2) If m≥(1/2)T, an optimal solution for Problem 2.1 is to make sure that the left side and the right side of every zero in θ is one. For instance, if T=10,m=7, then the sequences in Table 1 are two examples of the optimal scheduling.*


**Proof** **of Theorem 1.**First, we will show this problem is virtually the dual problem of Problem 3.1 [18] (in [18], the optimal scheduling for the n>(1/2)T case had a minor mistake, and here, we use the corrected format). Problem 3.1 in [18] studied the case when a perfect wireless network suffered from *u* times of DoS attack over a time window *T*. It was assumed that if there was no DoS attack, the sensor data packet would arrive at the remote estimator successfully; but if the attacker launched an attack, the packet would drop with probability $\alpha^{*}$. Then, the author denoted $\gamma_{k}=0$ or 1 as the attacker’s decision variable at time *k*, i.e.,
$$\gamma_{k}=\left\{\begin{array}{cc} 1, & \mathrm{if}\;\mathrm{an}\;\mathrm{attack}\;\mathrm{is}\;\mathrm{launched}\;\mathrm{at}\;\mathrm{time}\;k\\ 0, & \mathrm{otherwise}, \end{array}\right.$$
$$\mathrm{Packet}\;\mathrm{drop}\;\mathrm{rate}\alpha_{k}^{*}=\left\{\begin{array}{cc} 0, & \mathrm{if}\gamma_{k}=0\\ \alpha^{*}, & \mathrm{if}\gamma_{k}=1. \end{array}\right.$$By comparing the definitions of $\gamma_{k}$ and $\alpha_{k}^{*}$ with the definitions of $\lambda_{k}$ and $\alpha_{k}$, we can see that if we let m=T−u and change $\gamma_{k}=1$ into $\lambda_{k}=0$, as well as $\gamma_{k}=0$ into $\lambda_{k}=1$, then the solution to Problem 3.1 in [18] is exactly the solution to Problem 1, which completes the proof. û

### 2.2. The Case with Relay Nodes

In this part, we first show how the model in Figure 1 updates its state and error covariance. Next, we introduce the optimal scheduling for this model using the “converted table” method when the packet drop rate is one. Then, we show the way of generalizing this scheduling method for the α=1 case into the 0<α≤1 case using an algorithm, which generates the optimal scheduling for our model with relay nodes.

First of all, due to the assumption we made in the Introduction part, the way that the relay nodes receive data packets is the same as the estimator does, and the way they send the packets to the next node is the same as the sensor does. Furthermore, whether the node is sending low or high power, the definition of the scheduling variable $\lambda_{k}$ and the value of the packet drop rate remain unchanged in the relaying process. Therefore, the only difference between the relay nodes and the sensor is the way they update the estimation. For the relay nodes, the state estimation ${\hat{x}}_{k,i}$ and the error covariance $P_{k,i}$ are updated as follows [24]:(5)$$\begin{array}{c} \left({\hat{x}}_{k,i+1},\;P_{k,i+1}\right)=\left\{\begin{array}{cc} ({\hat{x}}_{k,i}^{s},\;P_{k,i}), & \mathrm{if}\;\mathrm{the}\;\mathrm{packet}\;\mathrm{arrives}\\ (A{\hat{x}}_{k,i},\;h\left(P_{k,i}\right)), & \mathrm{otherwise}. \end{array}\right. \end{array}$$
where $P_{0,i}=P_{0,0}=\bar{P}$, for ∀i∈[2,n−1].

**Remark** **1.**
*In fact, k refers to the $k^{\mathrm{th}}$ signal the sensor has sent, so in this update, k does not change with the increase of i. The reason we call k the “time step” is to follow tradition of previous work such as [18]. Therefore, we can ignore the time delay caused by transmitting data from one node to another and apply (Equation 5) to describe the updates between relay nodes.*


Besides, according to (Equation 5), we obtain the expression of the expectation for the node’s error covariance when zero is sent at time *k*:(6)$$\begin{array}{c} E\left[P_{k,i+1}\right]=\left(1-\alpha \right)E\left[P_{k,i}\right]+\alpha h\left(E\left[P_{k,i}\right]\right). \end{array}$$

Thus:(7)$$\begin{array}{c} E\left[P_{k,i+1}\right]-E\left[P_{k,i}\right]=\alpha \left\{h\left(E\left[P_{k,i}\right]\right)-E\left[P_{k,i}\right]\right\}\geq 0. \end{array}$$

Hence, we can see that for relay nodes, when the high power is sent, the average error covariance of the following node does not change; and when the low power is sent, the average error covariance increases, as is shown in (Equation 7). Now, we consider two cases while searching for the optimal scheduling, that is the α=1 case and the 0<α<1 case, which is an extension of the α=1 case.

#### 2.2.1. Case 1: The Optimal Scheduling When α=1

Now, we show how to minimize the average error covariance of the estimator in Figure 1, that is how to solve the following problem:

**Problem** **2.**
$$\min_{\theta }Tr\left(J_{\left[n\right]}\left(\theta \right)\right)$$
$$s.t.\sum_{k=1}^{T}\lambda_{k,i}=m,\;i=0,1\cdots ,n-1.$$
*where $\lambda_{k,i}$ refers to the $i^{\mathrm{th}}$ node’s scheduling variable and $J_{\left[n\right]}$ is the estimator’s average error covariance when there are n−1 relay nodes functioning between the final estimator and the sensor (for example, in Figure 1, we have n=3) Next, we use a simple example to show the “converted table” method that helps solve this problem.*


**Example** **1.**
*Table 2 lists a strategy that is randomly chosen for each node in Figure 1 at every time step when T=6 and m=2. With the assumption α=1, we can obtain the error covariances accordingly in Table 3.*

*Now, we can verify that in terms of the states’ error covariances, Table 2 equals Table 4 because they have the same $P_{k,i},i\geq 0$.*


Note that in Table 4, we no longer need to consider the update of (Equation 2) in $\theta_{0}$’s scheduling sequence, because the way of the sensor-to-relay-node update is converted into the way of the relay-node-to-relay-node update. Then, there is only one way of updating the left in this new table, since states updated by (Equation 2) are converted to states updated by (Equation 5) by replacing the sensor with a different number of imaginary relay nodes that use imaginary strategies $\theta_{-2}$ and $\theta_{-1}$. We used these auxiliaries because with only one way left to update the error covariance, we are now able to find the optimal sending strategy for our system. In the following, we give the definition of imaginary relay nodes and imaginary strategies.

**Definition** **1.**
*Imaginary relay nodes refer to nodes that do not exist physically, but are come up with to reduce the two ways of updating error covariances of nodes’ states into one. Their sending strategies, labeled as $\theta_{i},i<0$, are constructed to make sure after replacing the updating strategy of (Equation 2) with that of (Equation 5) that the physically-existed nodes hold the unchanged error covariances.*


Our goal is to achieve the minimum $Tr[J_{\left[n\right]}\left(\theta \right)]$. To solve this problem, we first assume that in $\theta_{0}$, the high power signal oneis distributed as uniformly as possible and then find the optimal $\theta_{1},\cdots ,\theta_{n-1}$ under this $\theta_{0}$ condition. Next, we prove that this $\theta_{0},\theta_{1},\cdots ,\theta_{n-1}$ combination is optimal. By Example 1, we have unified the way the error covariance evolves (This method now works when α=1. A more practical way that applies to 0<α<1 will be given later.). $T\times J_{\left[n\right]}\left(\theta \right)$ is the sum of $P_{k,n}$, and the value of each $P_{k,n}$ only depends on the total number of zeros in the $k^{\mathrm{th}}$ column of the converted table above since ones do not change its value. For each column, the starting error covariance is $\bar{P}$, and for further analysis, we also need the following lemma:
**Lemma** **2** **([15]).***Let $\lambda_{min}\left(A^{\prime }A\right)$ be the minimal eigenvalue of $A^{\prime }A$, then when $\lambda_{min}\left(A^{\prime }A\right)\geq 1$ and t≥0, we have:*$$\begin{array}{c} Tr\left(h^{t+2}\left(\bar{P}\right)-h^{t+1}\left(\bar{P}\right)\right)\geq Tr\left(h^{t+1}\left(\bar{P}\right)-h^{t}\left(\bar{P}\right)\right), \end{array}$$
which leads to the following corollary:

**Corollary** **1.**
*When $\lambda_{min}\left(A^{\prime }A\right)\geq 1$, the node should send the m ones (high power signals) at the m time steps that have the m largest error covariances.*


From Lemma 2 and Corollary 1, we can obtain our core theorem:

**Theorem** **2.**
*When we have $\lambda_{min}\left(A^{\prime }A\right)\geq 1$ and α=1, the optimal offline strategy for each node is to make the sensor send $\theta_{0}$, which takes the form of the optimal sequences in Theorem 1, and make the rest of the relay nodes follow the strategy in Corollary 1.*


**Proof** **of Theorem 2.**To prove that the combination of Theorem 2 and Corollary 1 gives the optimal strategy, first we set up four sets, A, $A^{\prime }$, B, and $B^{\prime }$, where $A^{\prime }$ is the complement of A and $B^{\prime }$ is the complement of B. Then, if the scheduling sequence $\theta_{0}$ follows Theorem 2, we put it in the set A and represent their relationship as $\theta_{0}\in A$; otherwise, we put $\theta_{0}$ in $A^{\prime }$ and get $\theta_{0}\in A^{\prime }$. Next, if $\theta_{i}$ follows Corollary 1, we put it in set B and represent it as $\theta_{i}\in B,i\geq 1$; or else, we put it in set $B^{\prime }$ and get $\theta_{i}\in B^{\prime },i\geq 1$. From Lemma 2, we learn that when $\theta_{0}$ is fixed, the combination of $\theta_{0}\in A$ (or $\theta_{0}\in A^{\prime }$) and $\theta_{i}\in B,i\geq 1$ is better than that of $\theta_{0}\in A$ (or $\theta_{0}\in A^{\prime }$) and $\theta_{i}\in B^{\prime },i\geq 1$. Therefore, we only need to compare the combination of $\theta_{0}\in A$ and $\theta_{i}\in B,i\geq 1$ (or $\theta_{i}\in B^{\prime },i\geq 1$) versus the combination of $\theta_{0}\in A^{\prime }$ and $\theta_{0}\in B,i\geq 1$ (or $\theta_{i}\in B^{\prime },i\geq 1$) when each $\theta_{i},i\geq 1$ in B (or $B^{\prime }$) is fixed.The differences between the combination of $\theta_{0}\in A$ and $\theta_{i}\in B,i\geq 1$ and the combination of $\theta_{0}\in A^{\prime }$ and $\theta_{i}\in B,i\geq 1$ are:I. $\theta_{0}\in A^{\prime }$ will bring more zeros into the imaginary nodes’ scheduling $\theta_{i},i<0$;II. the number of zeros of $\theta_{0}\in A$ at each time step (i.e., the zeros of each column in the “converted table”) is the most evenly distributed among all possible scheduling sequences.Therefore,
(i)when we use $\theta_{0}\in A$ and $\theta_{i}\in B,i\geq 0$ to schedule the total communication energy in a time period at a length of *t*, we can assume that after applying the “converted table”, we get $n_{1,j}$ zeros (including the ones in $\theta_{i},i<0$) in each column of the table, and $\sum_{j=1}^{t}n_{1,j}=N_{1}$;(ii)when we use $\theta_{0}\in A^{\prime }$ for the scheduling and keep all the other conditions the same as those in (i), we will get $n_{2,j}$ zeros in each column, and $\sum_{j=1}^{t}n_{2,j}=N_{2}$;(iii)according to the previous analysis in I and II, we have $N_{1}<N_{2}$ and:
$$\sum_{j=1}^{t}\left|n_{1,j}-N_{1}/t\right|<\sum_{j=1}^{t}\left|n_{2,j}-N_{2}/t\right|;$$(iv)if we set $f\left(x\right)=Tr(h^{x}\left(\bar{P}\right))$, then f(x) is convex since we can verify that:
(8)$$\begin{array}{c} \frac{f\left(x_{3}\right)-f\left(x_{2}\right)}{x_{3}-x_{2}}\geq \frac{f\left(x_{2}\right)-f\left(x_{1}\right)}{x_{2}-x_{1}} \end{array}$$
for $x_{3}>x_{2}>x_{1},\forall x_{i}\in N$. Note that though f(x) is not continuous, its domain consists of discrete points that constitute a closed set. Therefore, on such a set, f(x) is still convex.(v)Then, the optimization of the combination of $\theta_{0}\in A$ and $\theta_{i}\in B,i\geq 1$ can be proven by using the following lemma on majorization theory:
**Lemma 3** **([4,29,30]).***For $x=\left(x_{1},x_{2},\cdots ,x_{n}\right)\in R^{n},y=\left(y_{1},y_{2},\cdots ,y_{n}\right)\in R^{n}$, we rearrange the order of the elements and put the $i^{\mathrm{th}}$ largest element as $x_{\left[i\right]}$, i.e.,*(9)$$\begin{array}{c} \left\{\begin{array}{c} x_{\left[1\right]}\geq x_{\left[2\right]}\geq \cdots \geq x_{\left[n\right]}\\ y_{\left[1\right]}\geq y_{\left[2\right]}\geq \cdots \geq y_{\left[n\right]}. \end{array}\right. \end{array}$$*if the following conditions hold,*(10)$$\begin{array}{c} \left\{\begin{array}{c} \sum_{i=1}^{k}x_{\left[i\right]}\leq \sum_{i=1}^{k}y_{\left[i\right]},k=1,2,\cdots ,n-1\\ \sum_{i=1}^{n}x_{\left[i\right]}=\sum_{i=1}^{n}y_{\left[i\right]}. \end{array}\right. \end{array}$$*then x is majorized by y.**What is more, x is majorized by y if and only if for all convex functions ϕ: R→R,*(11)$$\begin{array}{c} \sum_{i=1}^{n}\phi \left(x_{i}\right)\leq \sum_{i=1}^{n}\phi \left(y_{i}\right). \end{array}$$*When ϕ is non-decreasing, (Equation 10) can be relaxed into:*(12)$$\begin{array}{c} \sum_{i=1}^{k}x_{\left[i\right]}\leq \sum_{i=1}^{k}y_{\left[i\right]},k=1,2,\cdots ,n. \end{array}$$
(vi)if $N_{1}=N_{2}$ and $\sum_{j=1}^{t}|n_{1,j}-N_{1}/t|<\sum_{j=1}^{t}\left|n_{2,j}-N_{2}/t\right|$, then it is easy to verify that $n_{1,i}$ and $n_{2,i}$ have the relationship just as $x_{\left[i\right]}$ and $y_{\left[i\right]}$ in (Equation 10); but since what we have is $N_{1}<N_{2}$, $n_{1,i}$ and $n_{2,i}$ actually follow (Equation 12). Because $f\left(x\right)=Tr(h^{x}\left(\bar{P}\right))$ is convex, from (Equation 12) in (v) we get:
(13)$$\begin{array}{c} \sum_{i=1}^{t}f\left(n_{1,i}\right)\leq \sum_{i=1}^{t}f\left(n_{2,i}\right), \end{array}$$
which implies the relationship of the two sets of strategies’ error covariance and thus completes the proof. ◻

**Remark** **2.**
*The case of comparing the combination $\theta_{0}\in A$ and $\theta_{i}\in B^{\prime },i\geq 1$ and the combination $\theta_{0}\in A^{\prime }$ and $\theta_{i}\in B^{\prime },i\geq 1$ is the same and thus omitted.*


#### 2.2.2. Case 2: The Optimal Scheduling Algorithm When 0<α<1

When 0<α<1, the sensor and the relay nodes no longer share the same error covariance update equation. Hence, if we still want to use the “converted table” method, we need to modify $\theta_{i},i<0$ in the converted table. First, we use Table 5 to indicate how the error covariance of each node evolves at different time step (when there are two relay nodes).

**Remark** **3.**
*The reason we list the k=0 column is to help develop the algorithm to decide the number of imaginary zeros, and the reasons and details will be discussed in the algorithms we are going to use.*


From (Equation 7), we see the expectation of the relay nodes’ error covariance increases with *i* monotonically, and from (Equation 3), we can also deduce that the sensor’s error covariance expectation increases with *k* monotonically. Hence, if we convert the sensor’s expectation of error covariance into the relay nodes’ by approximation and unify the two kinds of updates, which is shown in Algorithm 1 in detail, then the converted value is unique and still represents the relative magnitude of the original error covariance. We are able to use this kind of “converting” because we do not require the exact value of the error covariance while searching for the optimal strategy; instead, we only need to compare the relative magnitudes of different strategies and obtain the strategy that generates the smallest magnitude. In Algorithm 1, we will use the converted approximation to compare the number of zeros added in $\theta_{i},i<0$. After this approximation, the optimization of the 0<α<1 case is exactly the same as the α=1 case. Therefore, finally, we can get the optimal strategy using the method from Section 2.2.1.
**Algorithm 1** Optimal Offline Scheduling**Require:**   1:The time horizon, *T*;2:The total amount of the times of the high power sent by each node, *m*;**Ensure:**   3:The optimal sensor and relay nodes’ power scheduling $\theta^{*}=\left(\theta_{0}^{*},\cdots ,\theta_{i}^{*},\cdots ,\theta_{n-1}^{*}\right)$ that minimizes $Tr[J_{\left[n\right]}\left(\theta \right)]$ at the remote estimator;4:At time k=0 and node i=0, set $P_{0,0}=\bar{P}$, and we have $E\left[P_{0,0}\right]=\bar{P}$,$E\left[P_{0,0}^{mark}\right]=\bar{P}$;5: 6:**while** 
i≠n **do**7:    i=i+18:    $E\left[P_{0,i+1}^{mark}\right]=\left(1-\alpha \right)E\left[P_{0,i}^{mark}\right]+\alpha h\left(E\left[P_{0,i}^{mark}\right]\right)$.9:**end while**10:**while** 
k≠T
 **do**
11:    k=k+112:    $E\left[P_{k+1,0}\right]=\left(1-\alpha_{k}\right)\bar{P}+\alpha_{k}h\left(E\left[P_{k,0}\right]\right)$.;13:    For each $E[P_{k,0}]$, find i∈[0,n] such that $E\left[P_{0,i}^{mark}\right]\leq E\left[P_{k,0}\right]<E\left[P_{0,i+1}^{mark}\right]$;14:    **if**
$E\left[P_{1,i}^{mark}\right]=E\left[P_{k,1}\right]$
**then**15:        $e_{k,1}=i$;16:    **else**17:        $e_{k,1}=i+0.5$;18:    **end if**19:**end while**20:Set $S_{i}=\left\{e_{1,i},\cdots ,e_{T,i}\right\},i\in \left[0,n\right]$.21:**while** 
i≠n
 **do**22:    i=i+123:    Find the $\theta_{i}^{*}=\left(\lambda_{1,i}^{*},\cdots ,\lambda_{T,i}^{*}\right),i\in \left[0,n-1\right]$ that minimizes $\sum_{k=1}^{k=T}P_{k,i+1}$ (the minimization is shown in Algorithm 2);24:**end while**25:At time k=T, we obtain $\theta^{*}$.26:**return** 
$\theta^{*}$


We show our method in Algorithm 1 as a summary. To validate this algorithm, there are several points requiring further illustration. Line 8 computes the expectations of the error covariances $E[P_{1,i}^{mark}]$ for a sequence that is comprised of *T* zeros. Line 12 computes the expectations of the error covariances $E[P_{k,1}]$ for the optimal sequence whose ones and zeros are distributed as uniformly as possible as is described in Theorem 1, and these expectations are updated by (Equation 3). Line 10 to Line 19 give us the number of imaginary nodes we should add at different time steps *k*, i.e., $e_{k,1}$, to get the “converted table”. This conversion is achieved by comparing $E[P_{k,1}]$ and $E[P_{1,i}^{mark}]$. For each $E[P_{k,1}]$, Line 13 finds i∈[0,n] such that $E\left[P_{0,i}^{mark}\right]\leq E\left[P_{k,1}\right]<E\left[P_{0,i+1}^{mark}\right]$, and *i* is the imaginary number of zeros from $\theta_{i}$-s,i<0 added in the corresponding Timek column in the above tables. *i* only measures how the error covariances are rounded after the conversion. The reason for using Line 14 to Line 18 will be explained afterwards. Line 20 counts the total number of zeros each node has emitted until time *k*, including those from imaginary nodes. Line 21 to 24 are valid due to the fact that when 0<α<1, the conclusions of Lemma 2 and Corollary 1 still hold, because from (Equation 6) and (Equation 7), we have:(14)$$\begin{array}{cc} \; & Tr\left(E\left[P_{k,i+2}\right]-E\left[P_{k,i+1}\right]\right)-Tr\left(E\left[P_{k,i+1}\right]-E\left[P_{k,i}\right]\right)\\ = & Tr\left(\alpha \left\{h\left(E\left[P_{k,i+1}\right]\right)-E\left[P_{k,i+1}\right]\right\}\right)-Tr\left(\alpha \left\{h\left(E\left[P_{k,i}\right]\right)-E\left[P_{k,i}\right]\right\}\right)\\ = & \alpha Tr(h\left(E\left[P_{k,i+1}\right]\right)-h\left(E\left[P_{k,i}\right]\right)-Tr\left(E\left[P_{k,i+1}\right]-E\left[P_{k,i}\right]\right))\\ = & \alpha Tr\left(A\left(E\left[P_{k,i+1}\right]-E\left[P_{k,i}\right]\right)A^{\prime }\right)-Tr\left(E\left[P_{k,i+1}\right]-E\left[P_{k,i}\right]\right)\\ = & \alpha Tr(\left(A^{\prime }A-I\right)\left(E\left[P_{k,i+1}\right]-E\left[P_{k,i}\right]\right)). \end{array}$$

Thus, the rest of the analysis is the same as the case when α=1, and that is why we can use Line 21 to Line 24 in our strategy when 0<α<1. Line 23 is shown in Algorithm 2 in detail. What is more, now we explain the reason for using Line 14 to Line 18. Line 17 helps discriminate $e_{k,i}$, because in this way, if there exists $E[P_{k_{1},0}]$ and $E[P_{k_{2},0}]$$\left(k_{1}\neq k_{2}\right)$ that fall in the same $\left[E\left[P_{0,i}^{mark}\right],E\left[P_{0,i+1}^{mark}\right]\right]$ range, then the one that happens to equal $E[P_{0,i}^{mark}]$ will get the priority to have its $e_{k,i}$ enter the set $S^{\left[m\right]}$ in Algorithm 2.
**Algorithm 2** Line 23 in Algorithm 1.**Require:**   1:The time horizon, *T*;2:The total amount of the times of the high power sent by each node, *m*;**Ensure:**   3:Find the $\theta_{i}^{*}=\left(\lambda_{1,i}^{*},\cdots ,\lambda_{T,i}^{*}\right),i\in \left[0,n-1\right]$ that minimizes $\sum_{k=1}^{k=T}P_{k,i+1}$;4:Choose the *m* smallest $e_{k,i},k\in \left[1,T\right]$ in $S_{i}$, and put them in a new set $S^{\left[m\right]}$.5: 6:**for** k=1:T **do**7:    **if**
$e_{k,i}\in S^{\left[m\right]}$
**then**8:        $\lambda_{k,i+1}^{*}=1$9:        $e_{k,i+1}=e_{k,i}$10:    **else**11:        $\lambda_{k,i+1}^{*}=0$12:        $e_{k,i+1}=e_{k,i}+1$13:    **end if**14:**end for**15:**return** 
$\theta_{i}^{*}$


In summary, the correctness of our algorithms is described in the following way: Algorithm 2 is the core of Algorithm 1, and the majorization theory supports Algorithm 2. We can consider Algorithm 1 as the combination of the converted table method that converted the 0<α<1 case into the α=1 case and the sending strategy that is applied to the α=1 case in Section 2.2.1.

## 3. The Infinite Time and Energy Case

Now, we know that the updates of error covariances in our system can all be converted into the form described by (Equation 6), so a clearer discussion on the non-existence of optimization for the infinite time and infinite energy budget case for our system can be conveyed. Opposite to our conclusion, the authors found the optimal sensor scheduling strategy for the infinite case in the WSN with no relay nodes by using the steady-state analysis of the Markov chain in [7]. The reason is that we use (Equation 6) to update the error covariance, whereas they used (Equation 3). As a result, their states of the average error covariances were recurrent, yet our control for the nodes can only bring transient states for the covariance because the error covariance will only grow larger and never reduce. Hence, in our model, we cannot use any scheduling sequence that forms an optimal distribution or any distribution for the error covariance as a final stable state. Therefore, in this work, we only discuss the finite time and finite energy case.

## 4. Illustrative Examples

Here, we give the example for our models and algorithms. Assume the system parameters are defined as follows:$$A=\left(\begin{array}{cc} 2 & 0.6\\ 0.5 & 2 \end{array}\right),P=\left(\begin{array}{cc} 0 & 0\\ 0 & 0 \end{array}\right),$$
$$Q=\left(\begin{array}{cc} 0.3 & 0.3\\ 0.3 & 0.3 \end{array}\right),C=\left(\begin{array}{cc} 1 & 2 \end{array}\right),$$
R=1,T=6,m=2,n=3,α=0.9,
where *m* is the times of high power signals one node can send in Problem 2 and *n* is the number of nodes. Note that according to the previous theorems, *A* has to satisfy $\lambda_{min}\left(A^{\prime }A\right)\geq 1$. Then, with our algorithm, we get the optimal sensor and relay nodes’ scheduling sequences $\theta_{0}$, $\theta_{1}$, and $\theta_{2}$, respectively, in Table 6.

To show that our strategy is better than others, we first compared the error covariance of our strategy to the error covariances from another 10 randomly generated strategies. The results of the simulations are shown in Figure 2, and the implications of the labels are listed below:
The *y*-axis refers to $Tr[J_{\left[n\right]}\left(\theta \right)]$, and the *x*-axis implies the ordinal number of time the simulation runs.The circled line is the expected value of the average trace of the error covariance generated by our optimal strategy, and the dashed line that goes across it is the average value.The star-marked line is the value of the 10 average traces of the error covariance generated by random strategies, and the dashed line is their average value.

Next, we compared our optimal strategy with the strategy that every node was sending signals using their optimal stationary strategy mentioned in [2], i.e., the nodes all sent high power signals as uniformly as possible. Then, we obtained the following result in Figure 3:

The labels in Figure 3 are are as follows:The *y*-axis, the *x*-axis, and the circled line are the same as these in Figure 2.The dashed cyan line is the value of the average trace of the error covariance generated by the optimal stationary strategy.

From Figure 2, we can see that in this experiment, our strategy was always better than other strategies, which were picked randomly, and the trace of the average value of the error covariance of our strategy was also smaller. In Figure 3, we can see that the trace of the error covariance of the remote estimator was a fixed value when the sensor and the relay nodes all applied the optimal stationary scheduling to send the signals. The trace of the average value of the error covariance of our strategy was smaller than the optimal stationary strategy, so our strategy was better.

## 5. Conclusions and Future Work

We discussed the relay node structure for the sensor and relay nodes power scheduling problem that aims to optimize the remote state estimation under the communication energy constraint over a finite time horizon. We came up with the optimal scheduling strategy under the condition $\lambda_{min}\left(A^{\prime }A\right)\geq 1$ and proved the optimality of our scheduling strategy. Examples and simulations were provided to show that this strategy had better performance than other strategies. In the future, we will try to extend our algorithm to an online strategy that could adjust its scheduling due to feedback from the remote estimator. Furthermore, it would be interesting to consider the case when the relay nodes cannot only pass on information to the next node, but also send signals to its farther neighbors.

## Figures and Tables

**Figure 1 sensors-20-01073-f001:**
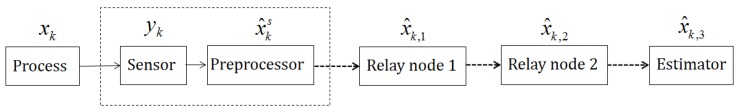
System structure diagram. The dashed-line arrows mark a network with packet drops.

**Figure 2 sensors-20-01073-f002:**
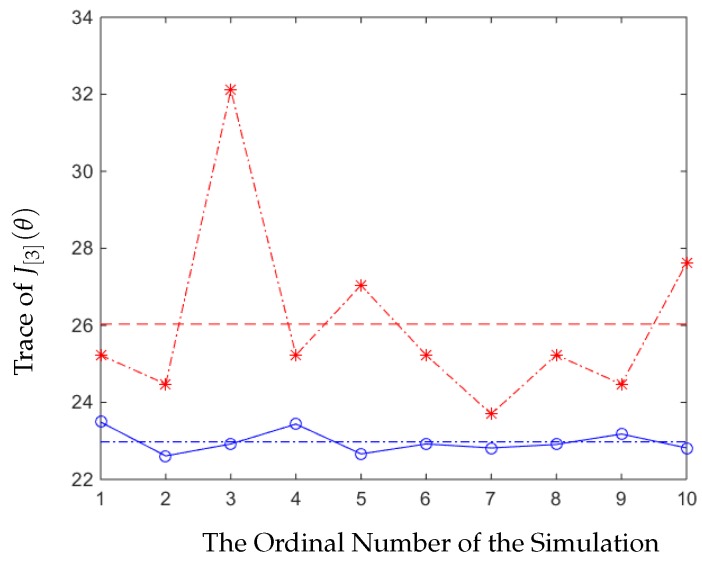
Comparing the error covariance of our strategy with the random strategies.

**Figure 3 sensors-20-01073-f003:**
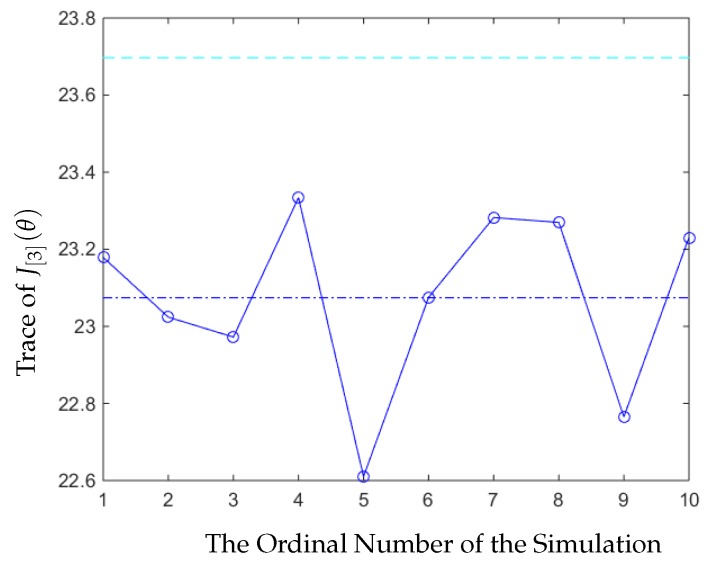
Comparing the error covariance of our strategy with the optimal stationary strategy.

**Table 1 sensors-20-01073-t001:** Two optimal scheduling sequences for Theorem 1.

1	1	0	1	0	1	1	1	0	1
1	0	1	1	1	1	0	1	0	1

**Table 2 sensors-20-01073-t002:** Strategy of each node at each time step.

Time k	1	2	3	4	5	6
$\theta_{0}$	0	0	1	0	1	0
$\theta_{1}$	0	0	1	0	1	0
$\theta_{2}$	0	0	1	0	1	0

**Table 3 sensors-20-01073-t003:** Error covariance of the strategy in Table 2.

Time k	1	2	3	4	5	6
$P_{k,1}$	$h(\bar{P})$	$h^{2}\left(\bar{P}\right)$	$\bar{P}$	$h(\bar{P})$	$\bar{P}$	$h(\bar{P})$
$P_{k,2}$	$h^{2}\left(\bar{P}\right)$	$h^{3}\left(\bar{P}\right)$	$\bar{P}$	$h^{2}\left(\bar{P}\right)$	$\bar{P}$	$h^{2}\left(\bar{P}\right)$
$P_{k,3}$	$h^{3}\left(\bar{P}\right)$	$h^{4}\left(\bar{P}\right)$	$\bar{P}$	$h^{3}\left(\bar{P}\right)$	$\bar{P}$	$h^{3}\left(\bar{P}\right)$

**Table 4 sensors-20-01073-t004:** “Converted table” for Table 2.

Time k	1	2	3	4	5	6
$\theta_{-2}$		1				
$\theta_{-1}$	1	0		1		1
$\theta_{0}$	0	0	1	0	1	0
$\theta_{1}$	0	0	1	0	1	0
$\theta_{2}$	0	0	1	0	1	0

**Table 5 sensors-20-01073-t005:** Error covariance of each node at each time step.

Time k	0	1	2	⋯	T
Node 0 (sensor)	$P_{0,0}$	$P_{1,0}$	$P_{2,0}$	⋯	$P_{T,0}$
Relay Node 1	$P_{0,1}$	$P_{1,1}$	$P_{2,1}$	⋯	$P_{T,1}$
Relay Node 2	$P_{0,2}$	$P_{1,2}$	$P_{2,2}$	⋯	$P_{T,2}$
Node 3 (estimator)	$P_{0,3}$	$P_{1,3}$	$P_{2,3}$	⋯	$P_{T,3}$

**Table 6 sensors-20-01073-t006:** Strategy of the sensor and two relay nodes in 6 time steps.

Time k	1	2	3	4	5	6
$\theta_{0}$	0	0	1	0	1	0
$\theta_{1}$	0	1	0	0	0	1
$\theta_{2}$	1	0	0	1	0	0

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
