# Peer review of "Optimal Sensor and Relay Nodes Power Scheduling for Remote State Estimation with Energy Constraint"

_sensors, 2020, doi:10.3390/s20041073_

Round 1

Reviewer 1 Report

This article studies the sensor and relay nodes power scheduling problem in a WSN.
The authors described the scope of their problem in Section 0.1 Backgound and Motivation, but it is not clear what motivate the authors to study.
It is recommended to clearly explain the motivation.
The author mentioned "the majorization theory" in the manuscript. It is recommended to add references.
The figure 1 mentioned in Section 0.1, but the description of notations described in Section 1. It is recommended to put the figure and the notation descriptions in the same section.
The title of Section 1.1 includes a verse, e.g. "When There is No Relay Node". It is recommended to make it short.
The authors described their proposed algorithm in Algorithm 1. It's recommended to show the correctness of the algorithm.
For the simulation, it is recommended to add the comparative results if applicable.

Reviewer 2 Report

The manuscript designed an algorithm for a special kind of sensor power scheduling problem. It introduced a sensor and relay nodes scheduling algorithm and proved its superiority. However, some parts need to be improved before it could be accepted for publication.
Major comments:
The definition for the initial value of the error covariance is not clear. In Page 4 Line 136, the author only gives the time-wise initial value of the error covariance. However, according to the rest of the paper, this time-wise initial value of the error covariance is also node-wise initial value. Please further explain the initial value of the error covariance and why it is initialized this way.
Minor comments:
Page 3 Line 103, there is an extra space in the sentence;
Page 3 Line 116, the notation for λ as the eigenvalue is not used in the rest of the paper. Instead, in the following λ refers to the scheduling variable, so the notation here should be deleted;
Page 4 Line 141, the footnote1 should be a note;
Page 5 Line 157, the footnote2 should be labeled as a reference;
Page 9 Line 213, the footnote6 and footnote7 should be a note and a remark, respectively.

Round 2

Reviewer 1 Report

No more comment.
